# Clinical and Genetic Features of *NR2E3*-Associated Retinopathy: A Report of Eight Families with a Longitudinal Study and Literature Review

**DOI:** 10.3390/genes14081525

**Published:** 2023-07-26

**Authors:** Sainan Xiao, Zhen Yi, Xueshan Xiao, Shiqiang Li, Xiaoyun Jia, Ping Lian, Wenmin Sun, Panfeng Wang, Lin Lu, Qingjiong Zhang

**Affiliations:** State Key Laboratory of Ophthalmology, Zhongshan Ophthalmic Center, Sun Yat-sen University, Guangdong Provincial Key Laboratory of Ophthalmology and Visual Science, Guangzhou 510060, China; xiaosainan@gzzoc.com (S.X.); yizhen@gzzoc.com (Z.Y.); xiaoxueshan@gzzoc.com (X.X.); lishiqiang@gzzoc.com (S.L.); jiaxiaoyun@gzzoc.com (X.J.); lianping@gzzoc.com (P.L.); sunwenmin@gzzoc.com (W.S.); wangpf7@mail.sysu.edu.cn (P.W.); lulin@gzzoc.com (L.L.)

**Keywords:** genotype–phenotype correlation, inherited retinal dystrophy, macular schisis-like change, *NR2E3*-associated recessive retinopathy, *NR2E3*-associated dominant retinopathy, retinitis pigmentosa

## Abstract

(1) Background: *NR2E3* encodes a nuclear receptor transcription factor that is considered to promote cell differentiation, affect retinal development, and regulate phototransduction in rods and cones. This study aimed to analyze the clinical characteristics and observe the prognosis of autosomal dominant retinopathy (ADRP) and autosomal recessive retinopathy (ARRP) associated with *NR2E3*; (2) Methods: *NR2E3* variants were collected from our exome sequencing data and identified per the American College of Medical Genetics and Genomics criteria. Data from our cohort and a systemic literature review were conducted to explore the *NR2E3* variants spectrum and potential genotype-phenotype correlations; (3) Results: Nine pathogenic variants/likely pathogenic variants in *NR2E3*, including five novel variants, were detected in eight families (four each with ADRP and ARRP). Follow-up data showed schisis/atrophy in the macula and retinal degeneration initiation around the vascular arcades with differences in ADRP and ARRP. A systemic literature review indicated patients with ADRP presented better visual acuity (*p* < 0.01) and later onset age (*p* < 0.0001) than did those with ARRP; (4) Conclusions: Macular schisis and retinal degeneration around vascular arcades may present as the prognosis of *NR2E3*-retinopathy, dominant, or recessive. Our data might further enrich our understanding of *NR2E3* variants and associated inherited retinopathy.

## 1. Introduction

*NR2E3* (nuclear receptor subfamily 2 group E member 3, OMIM: 604485) encodes a nuclear receptor transcription factor expressed in the adult retinal outer nuclear layer [1,2]. The nuclear receptor transcription factor contains two functional domains: the DNA-binding domain (DBD) (residues 47–130) close to the N-terminus and the ligand-binding domain (LBD) (residues 221–410) at the C-terminus [3]. Two highly conserved zinc fingers in NR2E3 (residues Cys 46–Cys 67, Cys 83–Cys 103) are located in the DBD, which facilitates binding to target DNA sites [4]. NR2E3 is considered to promote cellullar differentiation, affect retinal development, and regulate phototransduction in rods and cones by cooperating with the “cone-rod box” (CRX) and “neural retinal leucine zipper” (NRL) [2,4].

Inherited retinal dystrophies associated with *NR2E3* are classified into autosomal recessive and autosomal dominant inherited retinopathies. The specific *NR2E3* variant, c.166G>A (p. Gly56Arg), is reported to be the second most common variant causing autosomal dominant retinopathy (ADRP) [5]. The characteristic retinal changes in ADRP associated with *NR2E3* include the progressive degeneration of photoreceptors (first rods, followed by cones), attenuated retinal arterioles, pigment degeneration in the mid-periphery, and waxy optic disc [4,6,7]. Autosomal recessive retinopathy (ARRP) associated with *NR2E3*, also known as enhanced S-cone syndrome [2], Goldmann–Favre syndrome [8], and clumped pigmentary degeneration [9], is a rare condition. Previous studies have summarized the clinical signs of ARRP associated with *NR2E3* as yellow-white dots, nummular pigmentation, and variable macular schisis [10]. In ARRP associated with *NR2E3*, S-cones are enhanced while L-cones and M-cones have degenerated [11,12]. Enlarged and delayed S-cone electroretinographic (ERG) responses are present [10,13,14].

However, similar clinical manifestations, such as macular schisis [15] and hyper-autofluorescent rings around the vascular arcades [7,13,16,17,18], have been reported to present in both ADRP and ARRP associated with *NR2E3*, despite the differing pathogeneses [4]. There are few studies describing the incidence of similar phenotypes among the two diseases. Moreover, there is a paucity of literatures on the disease course of *NR2E3* associated ADRP and ARRP. Evaluating the characteristic clinical changes, prognosis, and genotype-phenotype correlations will contribute markedly to our understanding of these inherited diseases. In the present study, we summarized the clinical characteristics and observed the prognosis of ADRP and ARRP associated with *NR2E3*. In addition, we analyzed the spectrum of *NR2E3* variants and clinical data based on a review of the relevant literature.

## 2. Materials and Methods

### 2.1. Probands and Family Members

Clinical data and peripheral venous blood samples were collected from probands as well as their available family members at the Pediatric and Genetic Clinic and its associated Genetic Diagnostic Lab, Zhongshan Ophthalmic Center, Guangzhou, China.

### 2.2. Bioinformatics Analysis of NR2E3 Variants

*NR2E3* variants were identified in our patients with genetic eye diseases by whole-exome sequencing (WES) or targeted exome sequencing (TES). The procedures of WES and TES are described in our previous study [19]. According to the American College of Medical Genetics and Genomics criteria [20], multistep bioinformatics analysis was performed to evaluate the pathogenicity of these *NR2E3* variants. The Human Genome Mutation Database (HGMD, http://www.hgmd.cf.ac.uk/ac/index.php (12 September 2022)) was also referenced in defining the pathogenicity of *NR2E3* variants. Based on the Genome Aggregation Database (gnomAD: http://gnomad.broadinstitute.org/ (12 September 2022)), common variants (with a minor allele frequency of 0.01) were excluded. Sanger sequencing and co-segregation analysis were performed to analyze pathogenic/likely pathogenic variants (PVs/LPVs) of *NR2E3*. Silico programs were used to predict pathogenicity. The pathogenicity of the collected missense variants was evaluated using PolyPhen-2 (http://genetics.bwh.harvard.edu/pph2/index.shtml (12 September 2022)), LRT (www.genetics.wustl.edu/jflab/lrt_query.html (accessed on 12 September 2022)), Fathmm (http://fathmm.biocompute.org.uk/fathmmMKL.htm (12 September 2022)), REVEL (https://sites.google.com/site/revel genomics/ (12 September 2022)), and CADD (https://cadd.gs.washington.edu/ (12 September 2022)). The pathogenic possibilities of splicing variants were predicted by the Berkeley Drosophila Genome Project (http://www.fruitfly.org/ (12 September 2022)). Clinical data were collected for phenotype analysis.

### 2.3. Phenotypic Analysis

The clinical data of all available patients and family members were retrieved and retrospectively analyzed. The data included disease age at disease onset, age of first visit and latest visit, best corrected visual acuity (BCVA), macular changes on optical coherence tomography (OCT), ERG, autofluorescence fundus photographs, and color fundus photographs as routine ophthalmic diagnosis procedures.

### 2.4. Literature Review of NR2E3 Variants

Available studies were extracted from PubMed (https://pubmed.ncbi.nlm.nih.gov/ (12 September 2022)) and the HGMD using the keyword “*NR2E3*”. The associated phenotypic data in the studies published in English were collected and analyzed.

### 2.5. Statistical Analysis

Continuous parameters were presented as the median (range), while categorical parameters were presented as numbers and percentages. The BCVA was converted to the log of the minimum angle of resolution (logMAR) for statistical analysis. Visual acuity of counting fingers, hand motions, light perception, and no light perception were assigned values of 2.6, 2.7, 2.8, and 2.9 logMAR, respectively [21]. The proportion of two groups of fundus changes was compared using the Chi-square test. Statistical significance was set as a *p*-value of 0.05.

## 3. Results

### 3.1. Molecular Analysis of NR2E3 in Our Cohort

Nine PVs/LPVs, including five novels, were detected in *NR2E3* of 11 patients from eight families in our cohort (Table 1, Figure 1). Of the nine, two were presented as heterozygous changes in four families with ADRP. Both of the two heterozygous variants were missense variants involving Gly56. Furthermore, seven variants were presented as biallelic changes in four families with ARRP. The PVs/LPVs presented as biallelic changes and consisted of four missense variants, one in frame deletion, one splicing donor defect, and one frameshift deletion. Except for the nine, no other potential PVs/LPVs were detected to contribute to the phenotype in the eight families by analyzing the whole exome sequencing data. 

### 3.2. Clinical Data and Follow-Up of the Patients with NR2E3 Associated ADRP in Our Cohort

Of the four unrelated families with ADRP, seven patients were confirmed to harbor the heterozygous *NR2E3* variant and were received ocular examination. Fundus imaging demonstrated retinal pigment epithelial degeneration originated from the peripapillary region and progressed along the vascular arcades to the mid-peripheral retina and macula (Figure 2, Table 2). Hyper-autofluorescent rings were observed around the boundary of retinal pigment epithelial degeneration, and hypo-autofluorescence was present in the degeneration region (Figure 2a–e). Hyper-autofluorescence in the macula was related to the residual outer layer of the macula on OCT images. ERG records matched typical RP characteristics (Appendix A).

Of the seven patients, six patients visited our clinic at least twice. The median age at the first visit was 36.0 (18.0–44.0) years and that at the latest visit was 40.5 (24.0–55.0) years. The median follow-up interval was 10.5 (3.0–13.0) years. The visual acuity at the first visit was 0.2 (0.0–2.0) logMAR and decreased to 0.6 (0.1–2.0) logMAR at the latest follow-up. Two patients (F1-IV:1 and F3-II:1) underwent glaucoma surgery because of uncontrollable intraocular pressure. One patient’s (F1-V:2) macula changed from tiny schisis cavities to apparent macular schisis-like changes. The maculae of two patients (F1-V:1 and F3-I:2) changed from normal-like maculae to schisis-like maculae. Schisis resolution followed by macular atrophy was present in patient F1-IV:1.

### 3.3. Follow-Up of Patients with NR2E3 Associated ARRP in Our Cohort

In total, there were four patients from four unrelated families with ARRP in our cohort. In the patients with *NR2E3*-associated ARRP, retinal pigment epithelial degeneration, schisis-like changes, and hyper-autofluorescent rings were observed around the vascular arcades (Figure 3). The retinal structures inside the vascular arcades were relatively preserved in the four patients. Specific S cone responds were detected in two patients (Appendix A).

Of the patients with ARRP, three underwent ophthalmic follow-up at least twice with a median interval of 4.0 (2.5–5.0) years (Figure 3d–g, Table 2). The BCVA of the right eye of F6-II:1 improved from 20/200 to 20/32 with resorption of the schisis cavities. The BCVA of F7-II:1 declined slightly (right eye: 20/40 to 20/50; left eye, 20/50 to 20/100) over five years. Marked progression of the epimacular membrane was noted in the left eye of F7-II:1. For patient F8-II:1, the visual acuity improved from 20/40 to 20/22 with the medical assistance of an amblyopia specialist. However, tiny schisis cavities were found on OCT imaging at the latest visit (Figure 3a). For the four patients with ARRP, the BCVA at the first visit was 0.4 (0.3–1.0) logMAR. The visual acuity was 0.3 (0.1–0.7) logMAR at the latest visit.

### 3.4. Genotype-Phenotype Correlation Analysis of NR2E3-Associated Retinopathy

Common phenotypes were present in both types of *NR2E3*-associated retinopathy. In our cohort, macular schisis-like changes, retinal pigment epithelial degeneration, and a hyper-autofluorescent ring around the vascular arcades may represent the progression of both dominant and recessive *NR2E3*-associated retinopathy (Figure 4a). The retinal pigment epithelial degeneration originated from the peripapillary region and progressed along the vascular arcades to mid-peripheral retina and macula in the early stages of ADRP. In the advanced stages of ADRP, macular schisis-like changes occurred along with peri-macular retinal atrophy. The hypo-autofluorescence corresponded to the area of outer retinal atrophy. However, the residual normal retinal region inside the vascular arcades was present in patients with ARRP associated with *NR2E3* in our cohort in both the schisis and post-schisis stages (Figure 4b). The autofluorescence corresponded to the area that was normal. The specific S-cone response was positive in patients with ARRP but negative in patients with ADRP. Residual cone responses were observed in the early stages of ADRP, but no detectable cone or rod responses in the early stages of ARRP (Figure 4c). 

### 3.5. Literatures Review of NR2E3

Overall, clinical data and genetic data were available in 51 ADRP patients from 13 unrelated families with heterozygous p.Gly56Arg variant (Appendix A) and for 145 patients from 131 unrelated ARRP families with at least two PVs/LPVs of *NR2E3* (Appendix A). Phenotype analysis, genetic spectrum, and phenotype-genotype relation exploration were performed based on the data from our cohort and previous studies (Appendix A).

For *NR2E3*-associated ADRP, only two heterozygous variants, i.e., c.166G>A and c.166G>C, were detected so far, and both were predicted to result in the same residue change (p.Gly56Arg) at protein level if translated. Of the two heterozygous variants, the c.166G>A was much more variant (88.2%, 15/17) (Figure 5a) as compared to c.166G>C (11.8%). For the 58 patients from the 17 families with ADRP, the median age of onset was 12.0 (0.0–40.0) years. The median BCVA was 0.2 (0.0–2.9) logMAR. Macular changes were classified as normal-like, macular schisis-like, or macular atrophy. Normal-like macula was observed in 55.9% (19/34) patients with ADRP, macular schisis-like changes were observed in 14.7% (5/34) patients, and macular atrophy was observed in 29.4% (10/34) patients. No significant difference was noted in the present age (*p* = 0.94) or BCVA (*p* = 0.99) among patients with ADRP with different macular phenotypes (*p* > 0.05) (Appendix A).

Clinical and genetic data from 149 patients from 135 unrelated ARRP families with at least two PVs/LPVs of *NR2E3* were analyzed (Appendix A). A total of four variants were summarized, including 59 variants from previous studies and five novel variants from our cohort were referred to in the present study (Appendix A). The 64 biallelic variants could be classified as missense (57.8%, 37/64), in-frame deletion (7.8%, 5/64), frameshift (14.1%, 9/64), splicing (12.5%, 8/64), and nonsense (7.8%, 5/64) (Figure 4b). Most of these variants were located in two domain regions, 39.1% (25/64) in the DBD and 42.2% (27/64) in the LBD. The c.119-2A>C variant was the most frequent one (29.3%, 79/270), while c.932G>A was the second most frequent one (20.3%, 55/270). Missense + missense was the most common genotype in ARRP patients with biallelic variants, accounting for 43.7% (59/135) (Figure 4c,d).

As for the phenotype in patients with ARRP, the median age of onset of the 149 patients was 4.0 (0–32.0) years. The median BCVA was 0.3 (0.0–2.8) logMAR. Macular schisis-like change was the most common macular phenotype, accounting for 49.4% (43/87). Patients with macular atrophy accounted for 19.5% (17/87). Normal maculae were noted in 27.5% (24/87) of patients with *NR2E3*-associated ARRP. There was no significant difference in the present ages among patients with macular schisis, macular atrophy, or normal maculae (*p* = 0.30, *p* > 0.05). However, a significant difference in BCVA was observed (*p* = 0.02, *p* < 0.05) (Appendix A). Unclassified maculopathy was present in 3.4% (3/87) of patients. 

Genotype–phenotype correlation analysis was performed based on autosomal dominant and recessive retinopathy associated with *NR2E3* (Appendix A). After correction for age, patients with autosomal dominant retinopathy presented better visual acuity than patients with autosomal recessive retinopathy (*p* = 0.002, *p* < 0.05). The age of onset in patients with ARRP was significantly earlier than that in patients with ADRP (*p* = 0.00001, *p* < 0.001). The frequency of maculopathy in patients with ADRP (44.2%, 15/34) was significantly lower than that in patients with ARRP (72.4%, 63/87) (*p* = 0.004, *p* < 0.01).

In patients with ARRP, genotype–phenotype correlation analysis was performed based on a combination pattern of different types of variants (Appendix A). No significant differences were presented in BCVA among patients with biallelic missense variants (logMAR, 0.3, 0.0–2.8), missense + truncation variants (logMAR, 0.3, 0.0–1.3), and biallelic truncation variants (logMAR, 0.5, 0.0–2.6) (*p* > 0.05). The onset age of patients with biallelic missense variants was older than that of patients with biallelic truncation variants (4.0 [0.0–25.0] years old vs. 2.0 [0.0~20.0] years) with a statistically significant difference (*p* = 0.03, *p* < 0.05). The onset age of patients with missense + truncation variants (8.0 [0.0–32.0] years) was older than that of the other two groups, but the difference was not statistically significant (*p* > 0.05).

## 4. Discussion

In our cohort of patients with *NR2E3*¬associated retinopathies, p.Gly56Arg was the only known variant associated with ADRP. Both the c.166G>A and c.166G>C variants detected in our cohort resulted in Gly56Arg at the protein level. Macular schisis-like changes, retinal pigment epithelial degeneration, and a hyper-autofluorescent ring around the vascular arcade may represent the progression of the advanced stage in ADRP and schisis stage in ARRP in our cohort. S-cone ERG and autofluorescence fundus photographs helped identify the two diseases. Macular schisis resorbed as the disease progressed without intervention in ARRP patients. In our systemic review of the literature, the phenotype of the autosomal dominant form was milder than that of the recessive form, with better visual acuity, later age at onset, and milder maculopathy in the former. The data obtained in this study will be of value for genetic counselling and potential therapy for patient management.

The p. Gly56Arg variant is located in a region between the second and third cysteines of the first zinc finger [4]. Previous studies suggested that the mutant protein interacts with CRX, acts as a competition for dimerization [22], and causes DNA-binding failure [23,24], and it has been confirmed to be the cause of 1–2% of ADRP cases [5]. For recessive retinopathy, pathogenic *NR2E3* variants, which were demonstrated to be enriched in the DBD and LBD [9], lead to transcription factor abnormalities and disrupt the differentiation and distribution of photoreceptor cells in the retina [3].

Retinal degeneration of the region surrounding the optic disc was considered a symptom of advanced-stage retinitis pigmentosa, and this phenomenon was related to the distribution of cones [25]. In ADRP, S-cones were not enhanced, and the progressive degeneration of rods occurred before that of cones [4,6]. Retinal pigment epithelial degeneration originated from the peripapillary region and progressed along the vascular arcades to the mid-peripheral retina and macula in ADRP associated with *NR2E3* in our cohort. This phenotype differed from previous reports on typical retinitis pigmentosa [25,26]. Our results may contribute to our understanding of ADRP associated with *NR2E3*.

For ARRP associated with *NR2E3*, the disease course is divided into three stages according to the phenotype: the pre-schisis stage, the schisis stage, and the post-schisis stage [27]. In the pre-schisis stage, the macula may be relatively normal, and the peripheral photoreceptors undergo progressive degeneration [27]. Mottled retinal pigment epithelium along the vascular arcades may also be observed in the pre-schisis stage [28]. In our cohort, patient F8 II:1 received the medical assistance of an amblyopia specialist in this stage, and the visual acuity improved. In the schisis stage, macular schisis developed. White dots, retinal pigmentary epithelial degeneration around the vascular arcades, and peripheral retinoschisis also occur in this stage [27,29]. Visual acuity may decline markedly with the appearance of the macular schisis [30]. The disease progressed to the final stage with the resolution of macular schisis either spontaneously or with medical intervention [29,31,32]. Visual acuity may improve, as with the patient F6 II:2 in our cohort. However, previous studies reported irreversible retinal dysfunction [27,33].

To our knowledge, this is the first longitudinal observation study of *NR2E3* retinopathy in a Chinese population. A progression pattern of *NR2E3*-associated ADRP was observed in our patients by using multimodal imaging. Maculopathy progression in *NR2E3*-associated retinopathy, including the dynamic progression of presentation, resorption, and atrophy, was observed based on multimodal retinal images in our cohort. Novel variants in the biallelic status expanded the *NR2E3* variant spectrum. The data obtained in this study would be of value for genetic counselling and potential disease management. The limitations of the study are the lack of electrophysiological follow-up records and parallel ultra-wide-field ophthalmoscope, which might be a common problem for the retrospective study. In addition, the sample size of each retinopathy cohort may hinder the detailed comparison of individual parameters. A multi-center prospective study is expected to further confirm our findings and expand additional novel clues for the genotype-phenotype correlation, especially *NR2E3*-specific early signs of retinal changes, the potential window of gene-replacement intervention, and the long-term outcomes of *NR2E3*-associated retinopathy.

## Figures and Tables

**Figure 1 genes-14-01525-f001:**
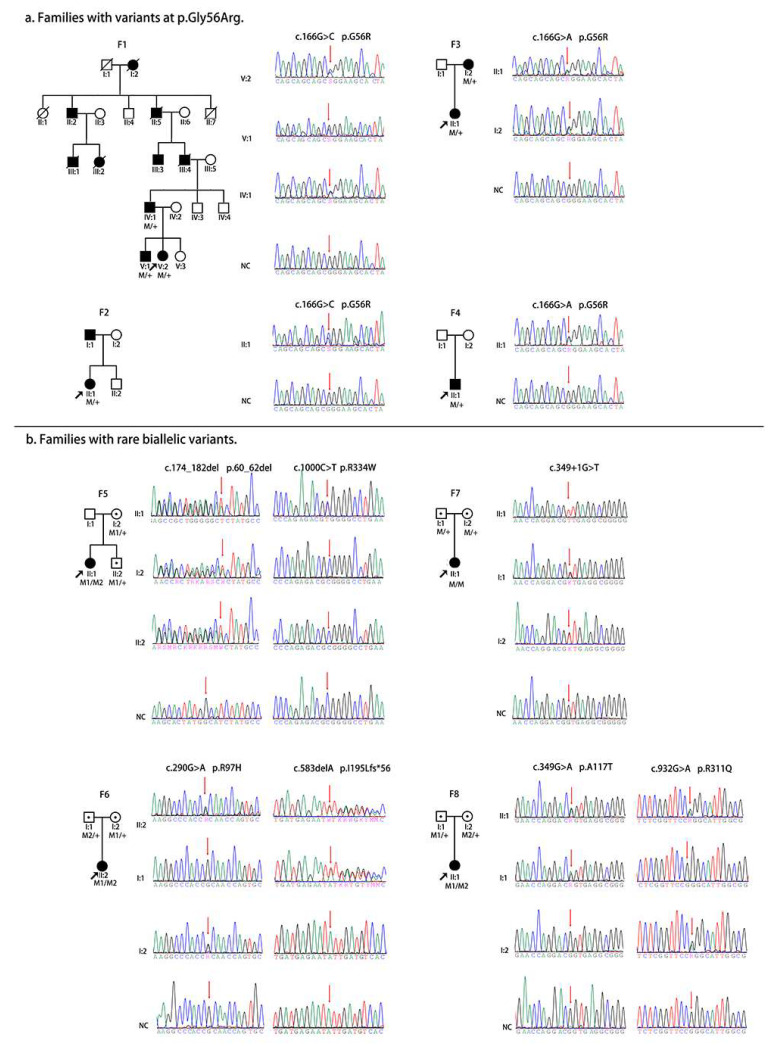
Pedigrees and sequence of eight families with rare *NR2E3* variants in our genetic database. There were four families with single heterozygous variants at p. Gly56Arg (**a**) and four families with rare biallelic variants in *NR2E3* (**b**). Family numbers are listed above the pedigrees. Sequences were listed to the right of the pedigrees.

**Figure 2 genes-14-01525-f002:**
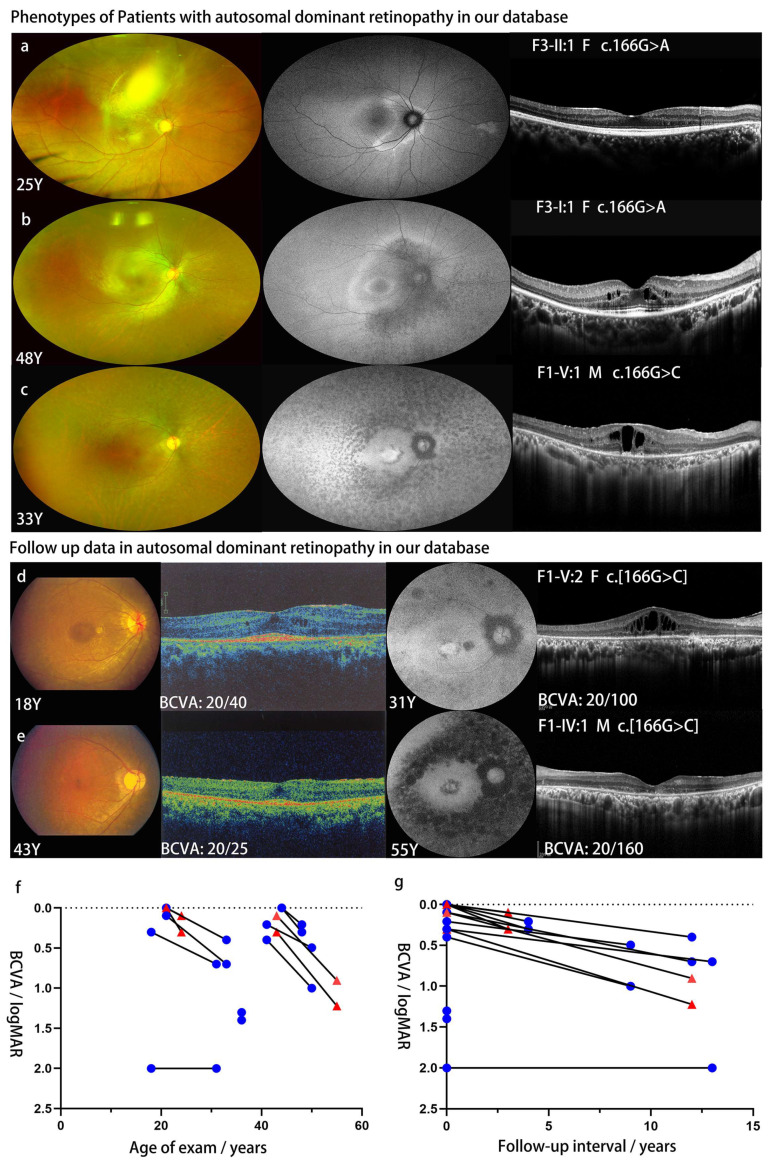
Multimodal fundus images of patients with single heterozygous variants p. Gly56Arg in *NR2E3* (**a**–**e**). In patient F3-II:1, a part of the retina was obscured by a residual cataract cortex (**a**). Plot showing BCVA (logMAR) of patients with ADRP in our cohort as a function of exam age (**f**) and length of follow-up time (**g**) per individual patient. Red triangles in (**f**,**g**) indicated the visual acuity of patients F3-II:1 and F1-IV:1, who underwent glaucoma surgery because of uncontrollable intraocular pressure. Blue circles in (**f,g**) showed the visual acuity of left patients without medical interruption.

**Figure 3 genes-14-01525-f003:**
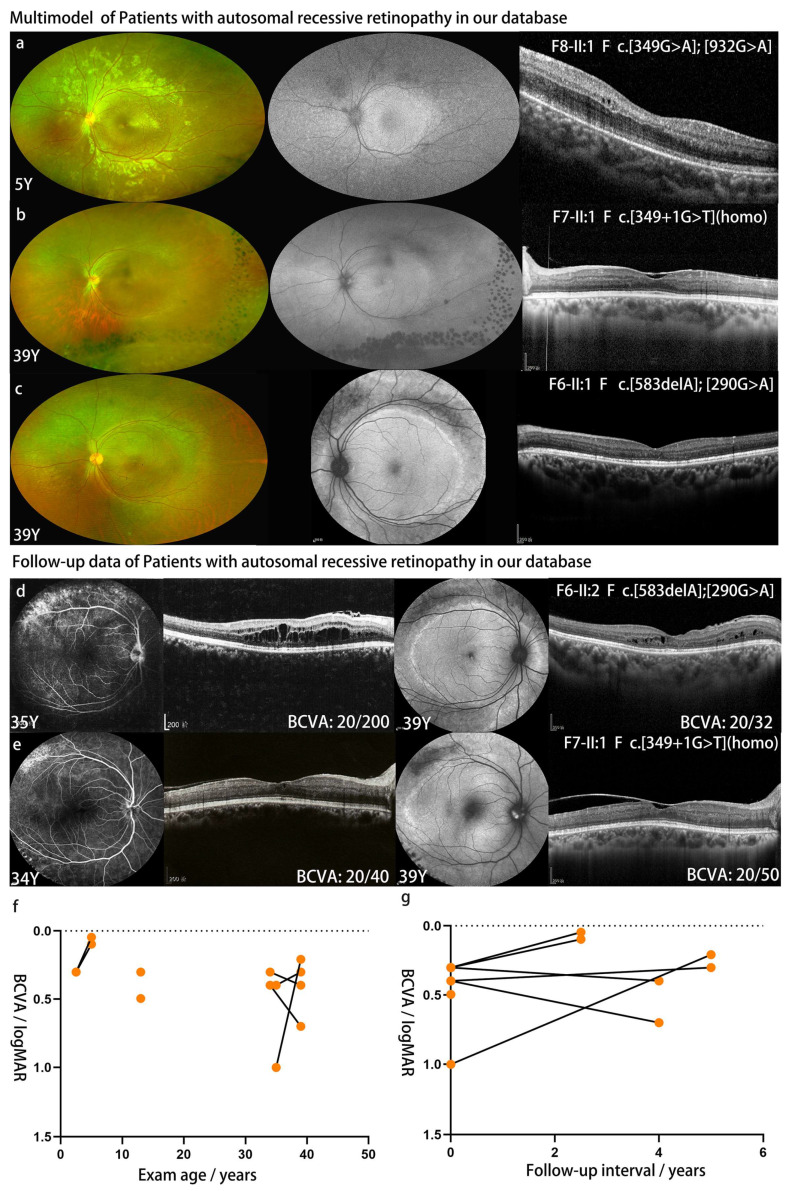
Retinal degeneration, schisis-like change and hyper-autofluorescence ring were observed around the vascular arcade in *NR2E3* ARRP patients (**a**–**e**). The BCVA of F6-II:1 improved from 20/200 to 20/32 with the resorption of schisis cavities (**d**). Marked progression of the epimacular membrane present in F7-II:1 and visual acuity declined slightly (**e**). Plot showing BCVA (logMAR) of patients with ARRP in our cohort as a function of exam age (**f**) and length of follow-up time (**g**) per individual patient.

**Figure 4 genes-14-01525-f004:**
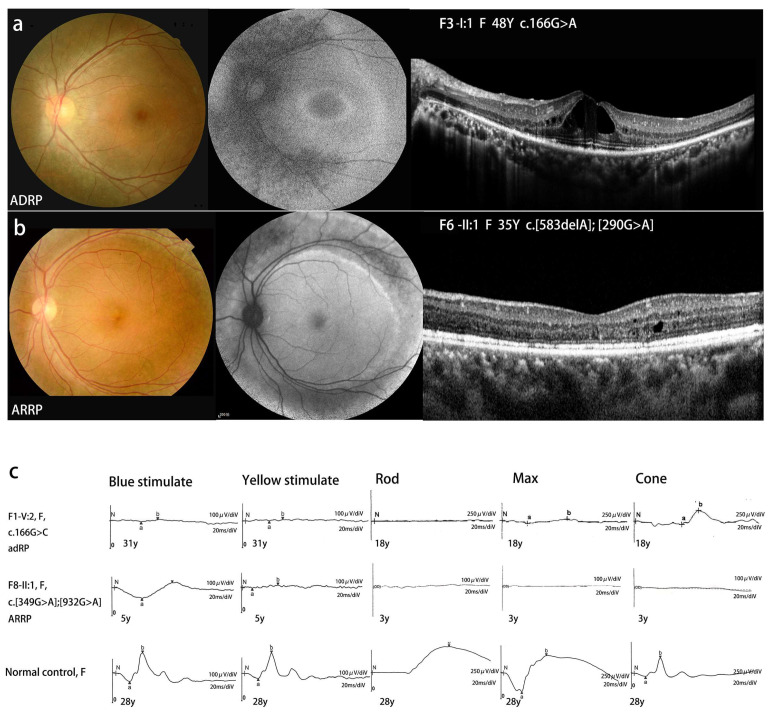
Comparison of patients with ADRP and ARRP in our cohort. The retinal pigment epithelial degeneration originated from the peripapillary region, a hyper-autofluorescent ring around the vascular arcades presented the progression of ADRP associated NR2E3 (**a**). A hyper-autofluorescent ring around the vascular arcades, and the residual normal retinal region inside the vascular arcades was present in patients with ARRP associated with NR2E3 in our cohort (**b**). The specific S-cone response was positive in patients with ARRP but negative in patients with ADRP (**c**).

**Figure 5 genes-14-01525-f005:**
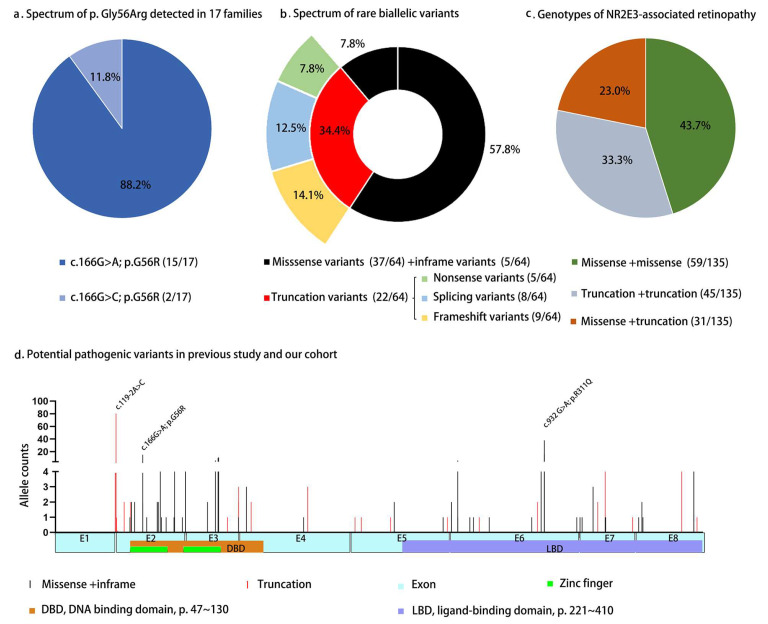
The p. Gly56Arg variant was detected in 17 families with *NR2E3*-associated ADRP, including four families in our cohort (**a**). The c.166G>A variant was detected in 15 families, and the c.166G>C variant was detected in two families. Totally, 64 variants associated with ARRP were identified, including five novel ones detected in our cohort (**b**). Genotypes of 135 families with biallelic *NR2E3* variants were summarized (**c**). The biallelic missense pattern accounted for almost half of the families (43.7%, 59/135). Distribution and frequency of PVs/LPVs in *NR2E3* (two variants associated with ADRP and 64 variants associated with ARRP) were presented (**d**). The variant of c.119-2A>C and c. 932G>A (p.R311Q) in *NR2E3* were the most common variants in families with biallelic *NR2E3* variants.

**Table 1 genes-14-01525-t001:** Nine variants in *NR2E3* were detected in our cohort.

No	Family	Position	Change	Effect	①	ACMG Evidence	②	③	④	⑤	⑥	⑦	⑧		HGMD
	ID	at Chr15	NM_014249	NM_014249									Total	EA	
**Dominant**															
1	F1, F2	72103870	c.166G>C	p.G56R	LPV	PS1, PP1, PP3, PP4	D*	D	D	0.57	16.00		0	0	DM
2	F3, F4	72103870	c.166G>A	p.G56R	PV	PS1, PS3, PM2, PP1, PP3, PP4	D*	D	D	0.57	15.66		1/152182	0	DM
**Recessive**															
3	F5	72103878	c.174_182del	p.60_62del	LPV	PM2, PM4, PP1, PP4	/	/	/	/	/		0	0	novel
4	F6	72104150	c.290G>A	p.R97H	LPV	PS1, PM2, PP3, PP4	D*	D	D	0.65	15.58		0	0	DM
6	F7	72104210	c.349+1G>T	SD	LPV	PM2, PM4, PP1, PP4	/	D	D	/	25.3	SSC	1/151650	0	novel
5	F8	72104209	c.349G>A	p.A117T	LPV	PM2, PP1, PP3, PP4	D*	D	D	0.59	16.00		0	0	novel
7	F6	72104687	c.583delA	p.I195Lfs*56	LPV	PM2, PM4, PP1, PP4	/		/	/	/		0	0	novel
8	F8	72105913	c.932G>A	p.R311Q	LPV	PS1, PP4, PP5	P	D	D	0.11	16.82		94/230880	2/17158	DM
9	F5	72106358	c.1000C>T	p.R334W	LPV	PM2, PM5, PP3, PP4	D*	D	D	0.5	17.08		35/275146	0	novel

Note: / = not applicable; ① = classification; ② = polyphen2; ③ = LRT; ④ = Fathmm-MKL; ⑤ = REVEL; ⑥ = CADD; ⑦ = BDGP; ⑧ = frequency in GnomAD; fs*= the frame shift ending at the stop codon; LPV = likely pathogenic variant; PS = pathogenicity with strong strength; PM = pathogenicity with moderate strength; PP = pathogenicity with supporting strength; D* = damaging; P = possibly damaging; D = deleterious; DM = disease-causing mutation; SSC = splice site changed; SD = splicing donor.

**Table 2 genes-14-01525-t002:** Clinical characteristic of *NR2E3* associated retinopathy in our cohort.

Family	Change	Effect	Gender	Age/Years			BCVA of FV		BCVA of LV		First	NB	Macular Change		Retinal Change
ID				Onset	FV	LV	OD	OS	OD	OS	Symptom		FV	LV	
**Dominant**															
F1-V:1	c.166G>C	p.G56R	F	ECH	18	31	20/40	20/2000	20/100	20/2000	NB	Y	MS	MS, MA	WD, MP-RPD
F1-IV:1	c.166G>C	p.G56R	M	NA	43	55	20/25	20/40	20/160	20/333	NA	Y	MS	MS, MA	MP-RPD
F1-V:1	c.166G>C	p.G56R	M	NA	21	33	20/25	20/20	20/100	20/50	NB	Y	NL	MS, MA	WD, MP-RPD
F2-II:1	c.166G>C	p.G56R	F	ECH	41	50	20/32	20/50	20/63	20/200	PV	Y	NA	MA	MP-RPD
F3-II:1	c.166G>A	p.G56R	F	21	21	24	20/20	20/20	20/40	20/25	high IOP	N	NL	NL	VA-RPD
F3-I:2	c.166G>A	p.G56R	F	NA	44	48	20/20	20/20	20/32	20/40	NB	Y	NL	MS	MP-RPD
F4-II:1	c.166G>A	p.G56R	M	28	36	NA	20/400	20/500	NA	NA	PV	Y	NA	NA	MP-RPD
**Recessive**															
F5-II:1	c.174_182del	p.60_62del	F	ECH	13	NA	20/63	20/40	NA	NA	PV	Y	MS	NA	VA-RPD
	c.1000C>T	p.R334W													
F6-II:1	c.290G>A	p.R97H]	F	32	35	39	20/200	20/50	20/32	20/40	PV	N	MS	MS	VA-RPD
	c.583delA	p.I195Lfs*56													
F7-II:1	c.349+1G>T (hom)	SD	F	ECH	34	39	20/40	20/50	20/50	20/100	NB	Y	MS	MS	PRS
F8-II:1	c.349G>A	p.A117T													
	c.932G>A	p.R311Q	F	ECH	2.5	5	20/40	20/40	20/22	20/25	PV	Y	NL	MS	VA-RPD

Abbreviations: fs*, the frame shift ending at the stop codon; F, female; M, male; ECH, early onset childhood; FV, first visit; LV, latest visit; BCVA, best corrected visual acuity OD, oculus dexter; OS, oculus sinister; NA, unavailable; IOP, intraocular pressure; PV, poor vision; NB, night blindness; Y, yes; N, no; MS, macular schisis-like; MA, macular atrophy; VA-RPD, pigmentary retinal degeneration; MP-RPD, mid-peripheral pigmentary retinal degeneration; WD, white dots; PRS, peripheral retinal schisis.

## Data Availability

The data presented in this study are available upon request from the corresponding author. The data are not publicly available due to ethical privacy.

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
