# Peer review of "Clinical and Genetic Features of NR2E3-Associated Retinopathy: A Report of Eight Families with a Longitudinal Study and Literature Review"

_genes, 2023, doi:10.3390/genes14081525_

Round 1

Reviewer 1 Report

This manuscript is well-written and well-structured and focuses on the description of clinical characteristics of ADRP in contrast to ARRP, their prognosis, and the incidence of similar phenotypes. The authors also analyzed the spectrum of NR2E3 variants and reviewed the most relevant literature. 

The experimental design is appropriate to test the hypothesis and is also supported by a statistical analysis of data. The results are very important for patient management and for genetics counseling. 

I only have few minor remarks and suggestions:

1)lines 81-82: explain better this sentence. The co-segregation with the disease could be used as a supporting information on the pathogenicity of a variant (criteria PP1 of ACMG). Explain the use of Sanger analysis for segregation, finding the variants only in the affected member of the family. 

2) all patients underwent WES or TES. Maybe it could be important to add information about other variants, double diagnosis, or also add a sentence like " no other variants associated with the phenotype were detected" in section 3.1, for example. 

3) lines 233-234: it does not clear the number of supplementary file that you referring to.

4) explain in the supplementary tables 1,2 and 3 the entity of "age" (onset/first visit): years? months? It's not very clear

Author Response

Reviwer 1

Comments and Suggestions for Authors

This manuscript is well-written and well-structured and focuses on the description of clinical characteristics of ADRP in contrast to ARRP, their prognosis, and the incidence of similar phenotypes. The authors also analyzed the spectrum of NR2E3 variants and reviewed the most relevant literature.

The experimental design is appropriate to test the hypothesis and is also supported by a statistical analysis of data. The results are very important for patient management and for genetics counseling.

Our response: Thank you for your kind suggestions.

I only have few minor remarks and suggestions:

1. lines 81-82: explain better this sentence. The co-segregation with the disease could be used as a supporting information on the pathogenicity of a variant (criteria PP1 of ACMG). Explain the use of Sanger analysis for segregation, finding the variants only in the affected member of the family.

Our response: Thank you for your careful reading. This was indeed a semantic error on our part. We changed the sentence from " Sanger sequencing and co-segregation analysis were performed to define pathogenic/likely pathogenic variants (PVs/LPVs) of NR2E3." to " Sanger sequencing and co-segregation analysis were performed to analyse pathogenic/likely pathogenic variants (PVs/LPVs) of NR2E3." in line 83.

2. all patients underwent WES or TES. Maybe it could be important to add information about other variants, double diagnosis, or also add a sentence like " no other variants associated with the phenotype were detected" in section 3.1, for example.

Our response: Thank you for your kind suggestions. Systemic analysis revealed no other pathogenic/likely pathogenic variants in the 11 patients from eight unrelated families. We added the statement " Except for the nine, no other potential PVs/LPVs were detected to contribute the phenotype in the eight families by analyzing the whole exome sequencing data. " in section 3.1 line 118.

3. lines 233-234: it does not clear the number of supplementary file that you referring to.

Our response: Thank you. We have added the correct supplementary material and number to the revised version of the paper.

4. explain in the supplementary tables 1,2 and 3 the entity of "age" (onset/first visit): years? months? It's not very clear

Our response: Thank you for your detailed reading. It was indeed an oversight on our part, we added the unit of age in years to the table.

Reviewer 2 Report

The authors sought to characterize the ophthalmic phenotype of ADRP and ARRP in a small chinese population with some correlation analyses done from similar databases from other races. This study is designed well and analysed sufficiently given the sample size of the current study. My concern is that the sample size is small and doesn't allow for proper sufficient analyses of observations. Further samples will be needed for sufficient statistical analyses. 

Some comments:

-Supplementary Table 1 should be included in the main manuscript as a proper description of the samples in the study. 

-Some indications in the OCT images depicting the ocular changes in the various figures. ERG can also be better marked for the changes in the graphs. 

Author Response

Reviwer 2

The authors sought to characterize the ophthalmic phenotype of ADRP and ARRP in a small chinese population with some correlation analyses done from similar databases from other races. This study is designed well and analysed sufficiently given the sample size of the current study. My concern is that the sample size is small and doesn't allow for proper sufficient analyses of observations. Further samples will be needed for sufficient statistical analyses.

Our response: Thank you very much for the advice.

Some comments:

1. Supplementary Table 1 should be included in the main manuscript as a proper description of the samples in the study.

Our response: Thank you for your kind suggestions. We added the Supplementary Table 1 in the main manuscript as your suggestion.

2. Some indications in the OCT images depicting the ocular changes in the various figures. ERG can also be better marked for the changes in the graphs.

Our response: Thank you very much! Yes, electrophysiological follow-up is important in this inherited retinopathy. However, in our cohort, the retrospective data for ERG were not as complete. We are unable to fix this problem at this time point due to the replacement of ERG recording facility from different sources during the long time period and the availability of follow-up visit at the expected times. We have included all the ERG data to in the supplementary figure S1 in the manuscript. This is the limitation we have. We have highlighted this in the limitations statement in the DISCUSSION section, line 319. Thank you again for advising.
